# Extremely cold ocean temperatures in iron formation brine pools of snowball Earth

Kai Lu[1,7], Lianjun Feng[2,3,7], Ross N. Mitchell [1,3] ✉, Maxwell A. Lechte[4] & Paul F. Hoffman[5,6]

For the severe low-latitude "snowball Earth" glaciations, glacial deposits occurring on all continents is well-established. However, cold, salty, ice-covered oceans—a salient prediction of snowball Earth—is difficult to establish geologically. Here we demonstrate that anomalously high iron isotope values ($\delta^{56}$Fe) of snowball iron formation—never observed in earlier anoxic Archaean oceans—can be attributed to additional temperature-dependent fractionation in extremely cold brine pools in the snowball ocean. Experiments and modeled fractionations relevant to the precipitation of iron formation demonstrate temperature-dependent $\delta^{56}$Fe fractionation, where colder temperatures correspond with more positive $\delta^{56}$Fe. Assuming the ~0.9‰ differential in $\delta^{56}$Fe values of snowball iron formation in excess of those preceding the Great Oxidation Event is due to temperature-dependent fractionation, we calculate that the temperature of the iron formation brine pools was −15 ± 7°C. Such cold snowball brine pools, colder than those in Antarctic margins today, represent Earth's coldest recorded ocean temperatures.

In addition to palaeomagnetic support for glacial deposits at low latitudes, snowball Earth is supported by geological evidence including glaciomarine sedimentary features, characteristic "cap carbonates", and the reprisal of widespread iron formation (IF) deposition since its disappearance in the Palaeoproterozoic Era[1–4] (Fig. 1). These Cryogenian iron formations (CIF) are mineralogically simple (predominantly composed of laminated hematite) and interbedded with glaciomarine deposits[5,6]. The occurrence of CIF during the Sturtian glaciation (ca. 717–660 Ma), the first of the two Neoproterozoic snowball Earth events, is attributed to the buildup of hydrothermally-derived ferrous iron ($Fe^{2+}$) while the sink for oxidation by oxygenic photosynthesis was cut off by the ice-covered ocean[1,7]. A combination of reduced dissolved sulfate in river water due to a weakened hydrological cycle[8] and mid-ocean ridges under less pressure due to sea-level fall (favoring a hydrothermal Fe>S ratio[9]) would have also tipped the relative balance in favor of Fe over S in the snowball ocean. Cyclostratigraphy constrains the duration of CIF deposition (for at least one well-studied unit, the Holowilena IF) to have lasted ~4 million years[6].

Iron isotope values ($\delta^{56}$Fe) of CIF have previously been exclusively interpreted in terms of redox variability[5,10]. According to this framework, the oxidation of $Fe^{2+}$ can result in the precipitation of Fe(III) (hydr)oxides that are enriched in isotopically heavy iron, and smaller degrees of partial oxidation (due to lower $O_2$ availability) along a distillation pathway produce more strongly positive $\delta^{56}$Fe values in sedimentary rocks[11,12]. Importantly, CIF $\delta^{56}$Fe values exhibit a wide range of variability only observed in IF deposited prior to the transition to an oxidizing atmosphere during the Great Oxidation Event (GOE) (Fig. 2), suggesting an important environmental perturbation. However, the fact that CIF $\delta^{56}$Fe values are anomalously positive even in comparison

[1]State Key Laboratory of Lithospheric and Environmental Coevolution, Institute of Geology and Geophysics, Chinese Academy of Sciences, Beijing, China. [2]Key Laboratory of Deep Petroleum Intelligent Exploration and Development, Center for Oil-Gas Theories and Methods, Institute of Geology and Geophysics, Chinese Academy of Sciences, Beijing, China. [3]College of Earth and Planetary Sciences, University of Chinese Academy of Sciences, Beijing, China. [4]School of Geography, Earth and Atmospheric Sciences, University of Melbourne, Melbourne, Australia. [5]Department of Earth & Planetary Sciences, Harvard University, Cambridge, USA. [6]School of Earth and Ocean Sciences, The University of Victoria, Victoria, Canada. [7]These authors contributed equally: Kai Lu, Lianjun Feng. ✉e-mail: ross.mitchell@mail.iggcas.ac.cn

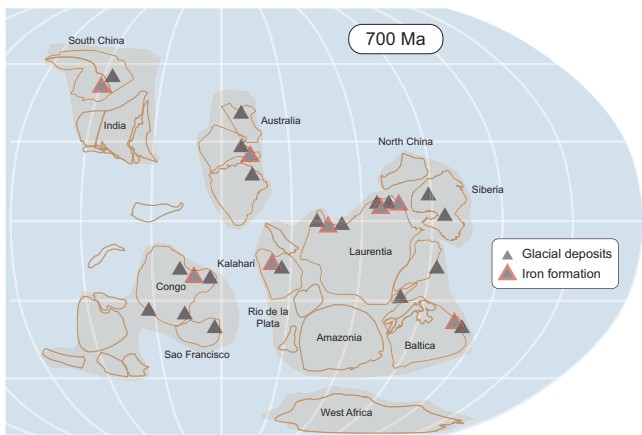

**Fig. 1 | Palaeogeographic distribution of Cryogenian iron formation during the Sturtian glaciation.** Palaeogeographic reconstruction and global CIF distribution modified from refs. 68,69.

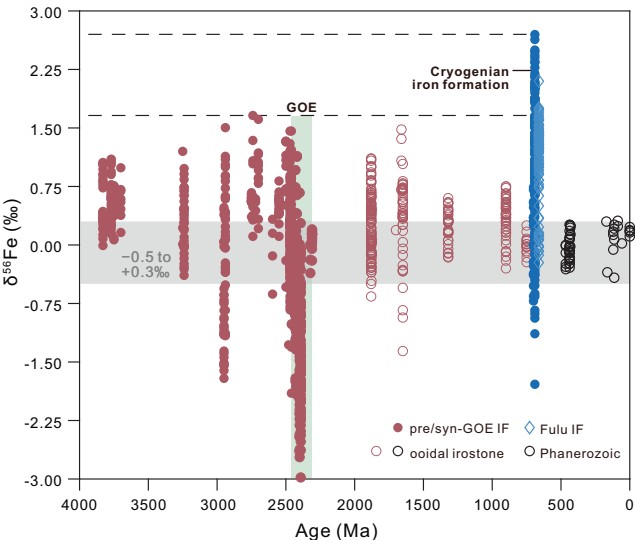

**Fig. 2 | Iron isotopes ($\delta^{56}$Fe) of iron formation (IF) and ooidal ironstone throughout Earth history.** Dashed lines indicate the anomalously positive $\delta^{56}$Fe values of CIF. The Fulu Formation of South China (light blue) may occur later in the Cryogenian than most CIF (dark blue; see text for details). Gray fields present the $\delta^{56}$Fe range of igneous and hydrothermal Fe(II) sources ($-0.5 < \delta^{56}$Fe $< +0.3$‰; ref. 11). GOE, Great Oxidation Event. Data provided in Supplementary Data 1. CIF, Cryogenian iron formation.

to IF preceding the GOE challenges the interpretation that the extreme fractionation can be attributed to redox conditions alone.

## Results

### Alternatives to redox fractionation

We consider several possible alternative explanations for redox. Given the large amount of time elapsed between the pre-GOE and Cryogenian intervals being compared, it is possible that a secular shift in the sources of Fe to the oceans between these two times could account for some or all of the 0.94‰ shift from pre-GOE IF to the values in CIF. Nonetheless, a secular trend analysis yields a line with essentially zero slope (Supplementary Fig. 1), allowing us to reject a change in Fe source in the oceans to explain the Cryogenian anomaly. It is also worth considering if the unique circumstances of the Cryogenian led to an unusual input of isotopically heavy Fe to the oceans at that time. Glacial erosion[13–15] presumably led to the mechanical breakdown and chemical dissolution of a continental crust that could have been covered by isotopically heavy Fe oxides produced by a billion years or more of chemical weathering under an at least mildly oxidizing atmosphere. Such a hypothesized biased detrital input, perhaps, could have led to an unusually heavy source of Fe to the oceans at this time. However, detailed geochemistry, rock magnetism, and petrology all consistently demonstrate that while there is a small detrital contribution from magnetite, the main source of Fe in CIF was mostly a hydrothermal-derived chemical precipitate preserved as authigenic hematite laths[6,16,17].

### Temperature-dependent iron isotope fractionation

In light of the uniquely positive CIF $\delta^{56}$Fe values that cannot be explained by redox nor other alternatives, we consider the temperature-dependent fractionation of $\delta^{56}$Fe associated with Fe (oxyhydr)oxide precipitation in cold, sub-ice shelf seawater. Arriving at a reliable fractionation factor is complicated by the possibility of kinetic isotope effects during Fe (oxyhydr)oxide precipitation, which will depend on a number of variables that control the rate and extent of Fe$^{2+}$ oxidation, and are difficult to disentangle from equilibrium isotope fractionation in natural systems[18]. Thus, for the purposes of our study, we assume that any kinetic effects are either negligible or can be assumed to be constant across the record. Figure 3a shows theoretical calculated results of the isotopic Rayleigh fractionation relevant to the precipitation of IF (Methods). Based on the observed anomaly in excess of pre-GOE values (Fig. 2), an increase in $\delta^{56}$Fe value of +0.94‰ extrapolates to a temperature of $-15.1 \pm 7$°C (Fig. 3b; Methods for uncertainty estimate).

We first consider for comparison any other additional empirical constraints on snowball seawater temperature potentially available.

Future work on CIF oxygen isotopes ($\delta^{18}$O) may provide an independent test, although the $\delta^{18}$O of hematite may be largely temperature-insensitive[19] and subglacial meltwaters could have had very large $\delta^{18}$O heterogeneity. Cryogenian constraints from temperature-sensitive oxygen "clumped" isotopes are already available; although this method has typically only been used on relatively much younger, much better preserved carbonates, every effort has been made in the Cryogenian to carefully analyze only the most pristinely preserved carbonate successions. Although the *absolute* temperatures are uncertain in this case, the *relative* temperature difference between glacial and preglacial temperatures is determinable and is $26 \pm 10$°C (ref. 20). Thus, if the temperature of the seawater during the Sturtian snowball was exactly $-15$°C as we have calculated, after adding the relative temperature difference 26°C to this, then the extrapolated pre-glacial temperature would have been 11°C. This estimate is actually quite close to the pre-glacial temperature of 9°C inferred from geological evidence in the form of the presence of temperature-sensitive ikaite[21,22], where a cold climate before snowball is also corroborated by independent constraints[23,24]. Also, even if the $\pm 10$°C uncertainty of the aforementioned clumped isotopes constraint is considered[20], along with the formation temperature of ikaite being $\leq 6$°C, our reconstructed glacial seawater temperature would still range from $-10$°C to $-30$°C. This is consistent with our estimate of glacial seawater temperature derived from iron isotopes, which falls between $-8$°C and $-22$°C.

## Discussion

The physical plausibility and implications of such a cold isotopically-inferred temperature estimate of $-15$°C for the seawater conditions associated with CIF sedimentation during snowball Earth should be considered. A major question to address is whether such cold seawater associated with CIF deposition is representative of the ambient snowball ocean or not. Our temperature estimate of $-15$°C clearly would require CIF seawater to be characterized by high salinity, arguably as high as >170 psu, considering the effect of salinity on freezing point[25]. We thus independently assess the salinity of CIF seawater using their Sr/Ba ratios (Methods), which most accurately

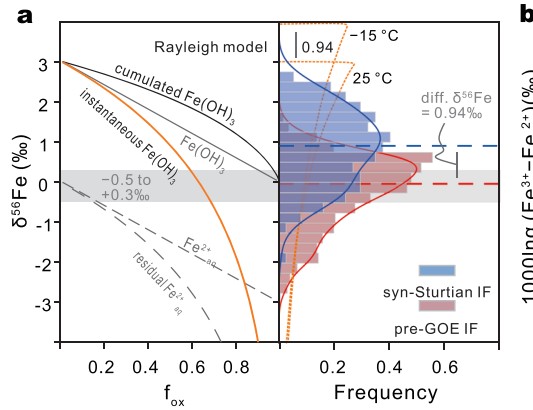
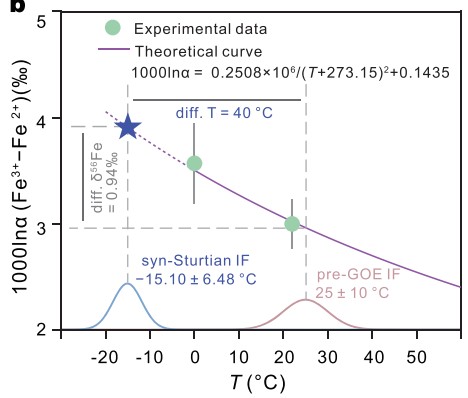

**Fig. 3 | Temperature estimate of the snowball ocean according to the CIF $\delta^{56}$Fe anomaly. a** Rayleigh distillation model of Fe isotope evolution during the genesis of IF and relative frequency distribution of $\delta^{56}$Fe values for pre-GOE IF and CIF (excluding data from the Fulu Fm). Orange dashed lines are theoretical $\delta^{56}$Fe distribution curves of instantaneous Fe (oxyhydr)oxide precipitation by progressive oxidation (distillation) at two different temperatures, assuming an initial source of $\delta^{56}$Fe = 0 (Methods). Median $\delta^{56}$Fe values for pre-GOE IF ($-0.03$, $n = 783$) and CIF (0.91, $n = 148$) are the red and blue dashed lines, respectively. A Mood test for equal medians yields a $p$ value of $8.5 \times 10^{-9}$, which is «<0.05, indicating that the null hypothesis can be rejected and that the two medians are strongly statistically distinct. **b** The temperature-dependent equilibrium fractionation between aqueous $Fe^{3+}$ and aqueous $Fe^{2+}$ (Methods). Dashed portion of the line is extrapolated to predict the snowball temperature based on the observed +0.94‰ $\delta^{56}$Fe anomaly (**a**). Experimental data[70] are consistent with the theoretical curve[59] used for extrapolation, and both are provided in Table 1. Bell curves indicate the $25 \pm 10\,°C$ Archaean seawater temperature used for the Gaussian-distributed Monte Carlo simulations spanning that range used to estimate uncertainty of the calculated snowball temperature (Methods). GOE, Great Oxidation Event. IF, iron formation. CIF, Cryogenian iron formation.

reflects Holocene records compared to other salinity proxies[26]. Compared to the salinity of the modern ocean, CIF seawater salinity would have been more than 4 times higher at approximately 150 psu (Fig. 4), which is broadly consistent with the high estimate implied by the $\delta^{56}$Fe-based temperature estimate. Stated conversely, under such extreme salinity conditions[25], the CIF seawater freezing point could have been as low as at least $-11°C$, which is, again, and as expected, broadly consistent with our $\delta^{56}$Fe-based temperature estimate. Thus, the elevated of salinity of CIF seawater provides independent support for the $\delta^{56}$Fe-based thermometry on CIF presented here.

Despite this consistency for CIF seawater temperature and salinity, it still does not address the question of whether such extremely cold and saline conditions were representative of ambient snowball seawater. First, it is worth considering that if the global snowball ocean had salinity levels of ~150 psu, it would require a > 75% reduction in volume compared to the modern ocean. Such a state, if realistic, would predict large-scale erosion of continental margins. However, while evidence of Cryogenian glacial erosion has been documented, it is highly spatially variable[13,15], and its scale is debated[14,27,28]. Furthermore, previous estimates of salinity levels during this period are thought to be only up to 70 psu (ref. 29), or even only ~50 psu by extrapolating the Phanerozoic salinity trend back to Cryogenian time (Supplementary Fig. 2). The Sr/Ba-based CIF seawater salinity estimate is thus more than two times higher, and therefore unlikely to represent ambient seawater.

It critical to note that, despite their widespread distribution globally (Fig. 1), CIF are generally thought to have been deposited in semi-restricted basins[17,30–32] (e.g., rift basins, glacial fjords, and overdeeps), so a substantial connection to the open snowball ocean may have been somewhat limited. As such, some modern ice shelves in Antarctica (e.g., Ross Ice Shelf) have a zone of basal freeze-on that is well seaward of the grounding zone, where there is strong basal melting[33,34]. In this sub-ice shelf circulation referred to as an ice pump[35–37], the freeze-on zone rejects salt that would sink, forming a bottom-water brine tongue (Fig. 5). Strong support for such a salinity-controlled freezing point depression comes from extreme temperatures as cold as $-13°C$ recorded in analogous Antarctic brines[38–42] (Fig. 5a). Notably, such sub-ice shelf brine pools, if an appropriate analog for CIF, do not preclude thriving microbial life in such cold and

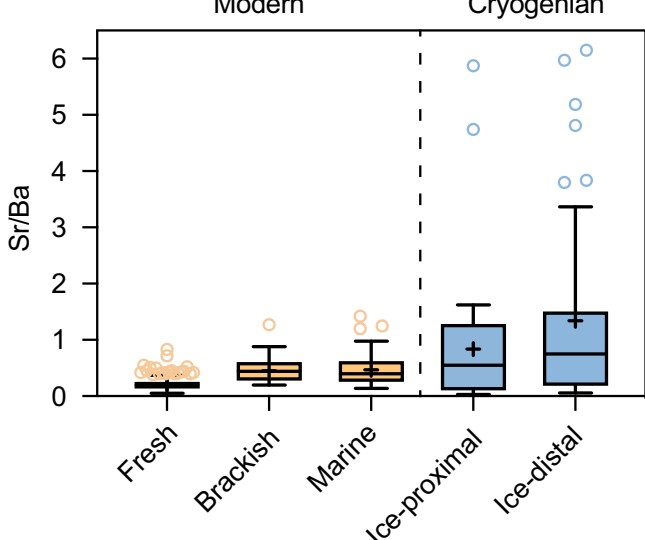

**Fig. 4 | CIF salinity compared to modern waters.** Tukey boxplot of Sr/Ba ratios for the modern sediments and CIF (Methods). Data for modern sediments are from ref. 67. Data for CIF are from ref. 5. Note that samples with high Ca content in modern sediments were excluded because of the substitution of Sr ions present in the water column for Ca ions in $CaCO_3$ (ref. 67). In addition, one Sr/Ba ratio (25.9) of one ice-distal CIF sample with an unusually high value was regarded as an outlier and removed. Horizontal lines within box plots represent median values. Crosses in the box plots represent mean values. CIF, Cryogenian iron formation.

salty conditions[39,43], as the specific CIF depositional setting has been proposed as a refugium for life during the severe snowball conditions owing to its delivery of oxidized subglacial meltwater potentially supporting marine aerobic environments at the ice ground line[5].

If such an ice shelf basal melting and refreezing salt expulsion zone was also characteristic of CIF sedimentation, the locally enhanced salinity could help account for both the very high salinity and cold seawater temperature recorded by the very positive $\delta^{56}$Fe values in CIF (Fig. 5b). Such a sub-shelf brine pool model has in fact been used to explain the apparent tendency of CIF to occupy depressions on the

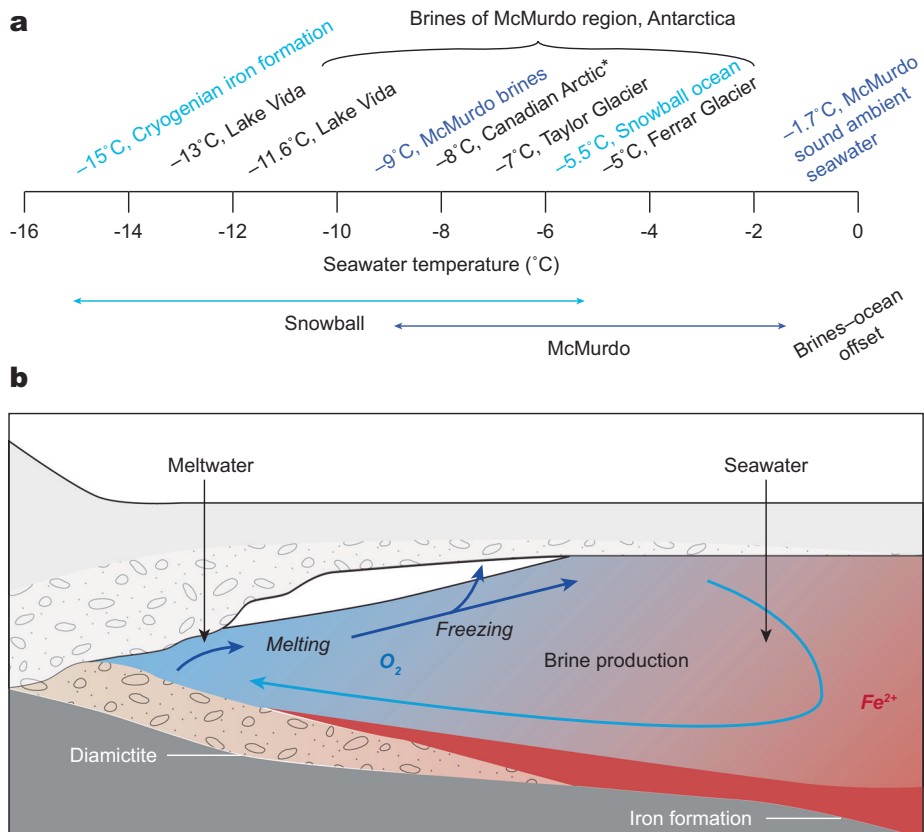

**Fig. 5 | Cryogenian iron formation brine pools of snowball Earth. a** Comparison of Cryogenian seawater estimates with those of modern Antarctica. The −15°C estimate for iron formation brines of snowball Earth (this study) displays a similar offset with the ambient snowball ocean[44] than brines and ambient seawater in modern Antarctica[38–42], albeit with temperatures shifted -5°C colder during snowball Earth than today. Estimate in blue for Antarctica is the mean of McMurdo brines. Note also one cold brine estimate included from the modern Canadian arctic[71]. Estimate for global snowball ocean is based on slight pressure-based freezing-point depression from the overlying sea glacier, under a range of salinities and water depths (or equivalent ice thicknesses)[44,72]. **b** Model for CIF deposition and brine production. Oxidation of the anoxic snowball ocean by oxidized meltwater pulses[5]. Brine production by salt expulsion during meltwater refreezing[30,35]. CIF, Cryogenian iron formation.

palaeo-seafloor[30]. Notably, comparison of our $\delta^{56}Fe$-based estimate of −15°C for cold CIF seawater appears to be offset from the ambient snowball ocean of −5.5°C, recently estimated according to freezing point depression due to the pressure imposed on the snowball ocean by the sea glacier[44]. The Cryogenian brine–ocean offset is similar to that of modern Antarctica (Fig. 5a), implying control by the similar basic ice physics mechanism of salt expulsion, only shifted to colder temperatures for the Cryogenian.

Such a model for the cold and saline CIF seawater, requiring basal melting, has also been used to explain the oxidation mechanism for CIF formation itself (Fig. 5b). A close association exists between CIF $\delta^{56}Fe$ values and glaciomarine setting, where sedimentation environment is controlled by the distance from the grounding line where the ice shelf begins to float[5]. The characteristically positive CIF $\delta^{56}Fe$ values are associated with ice-distal facies, whereas negative $\delta^{56}Fe$ values occur in ice-proximal facies, where such a $\delta^{56}Fe$ gradient have been interpreted to reflect oxidized meltwater mixing with the anoxic and ferruginous snowball ocean during glacial advance[5]. Interestingly, the $\delta^{56}Fe$ gradient observed in different glaciomarine facies with distance from the ice ground line is also in the right direction (i.e., more negative toward the grounding line) as that predicted by the temperature gradient. Seawater in the ice-proximal glaciomarine environment would have been warmer due to the supply of basal meltwater at the grounding line, and because of the lower salinity and concomitant freezing point increase in this setting. Thus, both redox-sensitive and temperature-dependent fractionations are consistent with each other, and it is likely that both contributed to the CIF $\delta^{56}Fe$ gradient. Comparison to analogous pre-GOE anoxic oceans (Fig. 2) argues, nevertheless, that the extreme snowball temperatures determined here are required to explain the full magnitude of the CIF $\delta^{56}Fe$ gradient.

Generally, most CIF sections have been shown to exhibit upsection increases in $\delta^{56}Fe$ (refs. 10,45). According to our results which predict temperature gradient-driven trends in Fe isotope signatures, CIF deposition therefore occurred during an interval of glacial advance—which had previously been ambiguous based on the unknown phasing of Milankovitch cycles identified in CIF modulating the snowball ice sheet[6]. The oxidizing agent for CIF deposition was heretofore enigmatic, with two hypotheses of opposite glacial phasing, either oxidation from (i) oxidized glacial meltwater reaching tidewater during glacial advance[5] or (ii) the oxidized atmosphere exchanging with seawater in polynyas during glacial retreat[1]. Increasing magnetic susceptibility associated with an increasing abundance of hematite that corresponds closely with increasing $\delta^{56}Fe$ values upsection[6] can therefore be attributed with colder temperatures of glacial advance. This collective observation provides support for a "hard" snowball Earth, where oxidation for CIF deposition only occurred from surges of oxidized continental-derived meltwater[5], but is unable to address the presence (or absence) of a "soft" snowball or 'waterbelt'[46,47].

## Methods

### Data compilation

Iron isotope ($\delta^{56}$Fe) data from iron formation (IF) and ooidal iron-stones were compiled from various data sources. Pre-1700 Ma $\delta^{56}$Fe data come from refs. [48,49]. Those authors include anomalously positive $\delta^{56}$Fe from the ca. 3.46 Ga Marble Bar Chert of the Pilbara craton, originally interpreted as a primary signature indicative of a strongly anoxic Archaean ocean[50]. However, detailed petrography of the hematite-bearing chert has since revealed the presence of secondary hematite replacement, questioning the reliability of those data as a primary oceanographic signature[51]. Because of this complication, and because we only include bulk-rock $\delta^{56}$Fe data from iron-rich chemical sedimentary rocks (i.e., IF and ooidal ironstones) so as to compare similar datasets, the Marble Bar Chert data are excluded here. For comparison, $\delta^{56}$Fe data for ooidal ironstones at various ages between ca. 2 and 0.7 Ga from ref. [52] are also shown. Cryogenian iron formation (CIF) $\delta^{56}$Fe data were compiled for the Chuos Formation of Namibia[5], the Fulu Formation[45,53–56] and Xiafang Formation[45] of South China, the Holowilena Ironstone of South Australia[5,10], and the Kingston Peak Formation (Death Valley)[57], the Rapitan Group (Northwest Territories, Canada)[58], and the Tantonduk Iron Formation (Yukon–Alaska)[10] of Laurentia.

### Temperature and uncertainty estimation

$\delta^{56}$Fe isotope data are used to infer temperature ranges of the snowball ocean. Of the CIF $\delta^{56}$Fe data, those from the Fulu Fm are excluded due to the lack of direct evidence of glaciation and the suggestion that it occurs later in the Cryogenian than most CIF[53]. The 0.94‰ $\delta^{56}$Fe difference between the median $\delta^{56}$Fe values of pre-GOE IF and the CIF (excluding Fulu Fm) (Fig. 3; Supplementary Data 1) is taken as the magnitude of the temperature-dependent fractionation (assuming that positive anomalies below this Cryogenian-only threshold can be attributed to redox-sensitive fractionation related to anoxia). Median values are used given that both datasets being compared have heavy tailed distributions. As the $\delta^{56}$Fe difference between the two groups would be larger if means were used, this approach is also conservative. Including the Fulu Fm would only increase the $\delta^{56}$Fe difference (Supplementary Fig. 2), so its exclusion is the more conservative approach, in addition to the ambiguities surrounding its age and syn-glacial or interglacial association[53].

The petrogenesis of IF follows a two-step pathway, i.e., the oxidation of aqueous $Fe^{2+}$ to aqueous $Fe^{3+}$ and precipitation of aqueous $Fe^{3+}$ to $Fe^{3+}$-(hydr)oxide[45]. Accordingly, the $\delta^{56}$Fe signature of IF involves a temperature-dependent equilibrium fractionation between aqueous $Fe^{2+}$ and aqueous $Fe^{3+}$ (refs. [59,60]) and a subsequent kinetic fractionation during precipitation of aqueous $Fe^{3+}$ to Fe (hydr)oxide[11]. The oxidation of aqueous $Fe^{2+}$ to aqueous $Fe^{3+}$ can reach isotopic equilibrium rapidly, resulting in enrichment of heavy iron isotope in aqueous $Fe^{3+}$ (ref. [60]). In the case of fast precipitation, the light iron isotope tend to be enriched in $Fe^{3+}$-(hydr)oxides[11]. Collectively, the observed $\delta^{56}$Fe values of Fe(oxyhydr)oxides could be approximately considered to be controlled by the temperature-dependent equilibrium fractionation between aqueous $Fe^{2+}$ and aqueous $Fe^{3+}$, and the fraction of Fe(II) oxidation and precipitation rate. Currently, nothing is known about possible Fe(II)/Fe(III) mass balance differences between the pre-GOE and Cryogenian conditions, and is thus possibly worthy of investigative testing.

The equally large spread in $\delta^{56}$Fe values of both the pre-GOE IF (4.64‰) and CIF (4.49‰) can be well explained by the Rayleigh distillation model[5,61] (Fig. 3a). The $\delta^{56}$Fe datasets with different aged intervals show distinct distributions (Supplementary Fig. 4). The dataset for the 3830–2310 Ma interval and CIF have the most similar frequency distributions and nearly identical $\delta^{56}$Fe ranges between minima and maxima (Fig. 3a; Supplementary Fig. 3). Although the petrogenesis of BIF is a two-stage process, assuming the precipitation rates at the time of IF in the two intervals (pre-GOE and Cryogenian) did not systematically differ, the systematic variation in Fe isotopes between these two intervals can be taken as to be derived mostly from temperature-dependent equilibrium fractionation. That is, the systematic increased $\delta^{56}$Fe values of the pre-GOE IF and CIF could mainly reflect the difference of temperature-dependent equilibrium between aqueous $Fe^{2+}$ and aqueous $Fe^{3+}$. Assuming that the ambient temperature at the time of the formation of the pre-GOE IF is 25°C ($\pm 10$°C; ref. [52]), the estimated temperature of the formation stage of the CIF is $-15.1 \pm 7$°C.

The largest source of uncertainty in our estimation is arguably the range of possible pre-GOE (largely Archaean) seawater temperatures that must be assumed for the relative temperature calculation. Recently, $^{18}$O/$^{16}$O isotopes of marine carbonates and cherts have suggested a very hot Archaean ocean ranging from 50–85°C (ref. [62]), or even hotter[63]. However, combined O and H isotopes suggest surface temperatures <40°C (ref. [64]) and O isotopes of marine phosphates produce upper limits of only 26–35°C (ref. [65]). Also, the repeated occurrence of glacial deposits at 3.5, 2.9, and 2.7 Ga are at odds with claims of a hot Archaean ocean and are more in line with the more temperature estimates[66], leading to the range of 25°C ($\pm 10$°C; ref. [52]) we employ for our calculation.

If one were to include not only the uncertainty of the mean ocean temperature of the pre-GOE IF but also the uncertainties of both the large spread in $\delta^{56}$Fe values of the pre-GOE IF and CIF, then

### Table 1 | Fe isotope fractionation data[a]

| | $T$ (°C) | $T$ (K) | $1000\ln\beta$ [Fe(H$_2$O)$_6^{3+}$)][b] | $1000\ln\beta$ [Fe(H$_2$O)$_6^{2+}$)][b] | $1000\ln\alpha$ (Fe$^{3+}$ -Fe$^{2+}$)[c] | $1\sigma$[d] | Reference |
|---|---|---|---|---|---|---|---|
| Theoretical data (Density functional theory estimation) | –0.15 | 273 | 9.488 | 6.013 | 3.475 | | 59 |
| | 24.85 | 298 | 8.070 | 5.095 | 2.975 | | 59 |
| | 49.85 | 323 | 6.943 | 4.370 | 2.573 | | 59 |
| | 99.85 | 373 | 5.291 | 3.314 | 1.977 | | 59 |
| | 199.85 | 473 | 3.354 | 2.088 | 1.266 | | 59 |
| | 299.85 | 573 | 2.310 | 1.433 | 0.877 | | 59 |
| Experimental data | 0 | 273.15 | | | 3.57 | 0.38 | 60 |
| | 22 | 295.15 | | | 3.00 | 0.23 | 60 |

[a]Data show the isotope fractionation values between Fe(III) and Fe(II) calculated using first-principles Density functional theory (DFT) method[59], as well as the experimental results between Fe(III) and Fe(II)[60].
[b]Logarithm of the reduced partition function, ln$\beta$ (‰), for the pair $^{56}$Fe-$^{54}$Fe of aqueous Fe(III) or Fe(II).
[c]Isotope fractionation value, as $1000\ln\alpha$ (‰), between the aqueous Fe(III) and Fe(II), is: $1000\ln\beta_{Fe(III)}$ - $1000\ln\beta_{Fe(II)}$ at a given temperature $T$. Under different temperature conditions, the relationship between isotope fractionation values and temperature can be expressed as: $1000\ln\alpha = A*10^6/T^2 + B$, where $T$ is in Kelvin. For the modeled data from Fujii et al.[59], A = 0.2508 and B = 0.1435 (Fig. 3b).
[d]Error of isotope fractionation values ($1000\ln\alpha$) obtained by experiments.

the resulting uncertainty is exceedingly large (>± 60°C), much of which is impossible parameter space for snowball exceeding freezing. However, based on both the similarity in their distributions (Supplementary Fig. 3) and the systematic increase in $\delta^{56}Fe$ values of the pre-GOE IF and CIF (Fig. 2), we regard it reasonable to consider the error of the mean ocean temperature of the pre-GOE IF as the main source of uncertainty. Based on the theoretical equation of Fe isotopic fractionation in Fig. 3b, the uncertainty of the mean ocean temperature of the pre-GOE IF (± 10°C) alone is thus propagated to derive the error of the estimated temperature of the formation stage of the CIF (± 6.48°C).

**Estimation of salinity**
To estimate Sturtian salinity in this study, we utilized the Sr/Ba ratio of sediments, a newly developed geochemical indicator for evaluating palaeoceanographic salinity[67]. We find that the Sr/Ba ratios of synglacial CIF have spatially similar distribution characteristics to the modern ocean (Fig. 4), i.e., the Sr/Ba ratios are significantly lower in areas with strong nearshore freshwater influence (Fig. 4). This feature is clearly consistent with the glacial meltwater model of the CIF[5], i.e., the median value of the Sr/Ba ratio is significantly lower at sites with low salinity where glacial meltwater input is high (ice-proximal) than at sites with low glacial meltwater input (ice-distal). Thus, the highest Sr/Ba value (6.2) in the CIF (Fig. 4) represents a Sr/Ba ratio that more closely represents marine sediments of this period. In contrast, modern marine sediments, which are less influenced by freshwater, have Sr/Ba ratios up to 1.4 (ref. 67) (Fig. 4).

It is unlikely that such a high Sr/Ba ratio in the CIF can be interpreted as a rapid expansion in the Sr reservoir due to weathering or hydrothermal sources of Sr, but is more likely to result from the thickening of sea ice during this period that led to the concentration of seawater. This also brought about a consequent increase in seawater salinity. Therefore, using the linear relationship that exists between Sr/Ba ratio and modern seawater salinity, we can estimate the degree of seawater concentration during the snowball Earth period, which gives us the approximate salinity (152 psu) of the seawater at that time.

## Data availability
Iron isotope data compilation is provided in Supplementary Data 1.

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

## Acknowledgements

The manuscript benefited from critical feedback from Ariel Anbar, Kurt Konhauser, and Xiangli Wang. Funding was provided by The National Key Research and Development Program of China 2023YFF0803600 (LJF), National Natural Science Foundation of China grant 42488201 (RNM), Strategy Priority Research Program (Category B) of Chinese Academy of Sciences XDB0710000 (RNM), Key Research Program of the Institute of Geology and Geophysics, Chinese Academy of Sciences grant IGGCAS-201905 (LJF, RNM), and the President's International Fellowship Initiative 2021FYC0002 (RNM).

## Author contributions

Conceptualization: P.F.H., M.A.L., R.N.M. Methodology: L.J.F., K.L. Data compilation: L.J.F., R.N.M., M.A.L. Data interpretation: L.J.F., K.L., R.N.M. Writing – original draft: R.N.M. Writing – review & editing: L.J.F., K.L., R.N.M., M.A.L., P.F.H.

## Competing interests

The authors declare that they have no competing interests.
