## [Transparent Peer Review file · Nature Communications]

Extremely cold ocean temperatures in iron formation brine pools of snowball Earth

Corresponding Author: Professor Ross Mitchell

Version 1:

Reviewer comments:

Reviewer #1

(Remarks to the Author)

I feel that the authors have done a good job at addressing my concerns. Although I would have liked to see bit more details about how the surges of oxidized glacial meltwater worked, it is a fair point that this topic has already been covered by the co-authors elsewhere. Thus, I am happy to accept the manuscript.

Reviewer #2

(Remarks to the Author)

This ms presents a very interesting observation that Fe isotope compositions of the Cryogenian BIF are among the heaviest in Earth's history. As isotope fractionation is negatively correlated with temperature in general, this observation would imply that the temperature for Cryogenian BIF is extremely low, and this is what are the authors meant. However, BIF precipitates from water, and the freezing point of water is almost constant. Therefore, the fundamental reasoning of the ms is wrong. thus rejection is strongly suggested.

Reviewer #3

(Remarks to the Author)

Reviewers' comments for revised manuscript entitled: "Extremely cold ocean temperatures during snowball Earth" by Lu et al. This study describes the use of temperature-dependent iron isotope fractionation of Cryogenian iron formations (CIF) to propose that the temperature of the Snowball oceans was $-15^{\circ}\text{C} \pm 7^{\circ}\text{C}$, and that marine ice was correspondingly very thick. The application of temperature-dependent Fe fractionation to ocean temperature reconstructions is of interest, especially for time periods when many of the proxies available for more recent Earth history cannot be applied. At the request of the Editor, we focused on the uncertainty and plausibility of the revised manuscript's temperature reconstruction, and the authors' responses to previous reviewers' remarks.

We disagree with the manuscript's overall conclusion that the reconstructed ocean temperatures apply to the global Cryogenian ocean. We do not dispute the temperature reconstructions; rather, we find that the framing of the environmental context of these results, and the subsequent ramifications for global snowball conditions, are not adequately supported by the arguments given. The claims made in this manuscript would benefit from greater exposition, perhaps in another journal conducive to longer manuscripts as suggested by previous Reviewer #2.

Line numbers refer to the revised manuscript with integrated edits.

CIF depositional settings. The authors note briefly (in lines 202-204) that CIF are associated with semi-restricted basins that may not have had a substantial connection to the open snowball ocean. The reviewers feel that it is more likely that the reconstructed temperatures are in fact capturing conditions in restricted basins. Note that the Holowilena Formation is found within Sturtian-age rift basins of the Adelaide Superbasin (see e.g. authors' Ref. 17; Lloyd et al., Geological Magazine (2023)), and Neoproterozoic IF in general are typically associated with rift settings (authors' Ref. 13). The authors dismiss a

scenario in which the waters of these basins may have become isolated from the global ocean by the overflow of land-based glaciers, leading to the significantly higher salinity suggested by the Sr/Ba data (authors' Ref. 5). The authors deem the isolation of Holowilena unlikely owing to the presence of submarine ridge volcanism (authors' Ref. 46). However, none of the large scale volcanic events examined in authors' Ref. 46 are in the geographic vicinity of Holowilena deposition. Furthermore, the authors have not addressed the possibility that smaller-scale hydrothermal vents supplying Fe could have been active within these rift basins, as they are e.g. in the modern Red Sea (van der Zwan et al., *Communications Earth & Environment* (2023)). Considering this omission of a more in-depth discussion, we argue that the authors do not sufficiently support their assertion that their findings are representative of the global ocean.

Sea water temperatures. The authors argue (lines 116-124) that the difference between their calculated glacial sea water temperature of -15°C and an extrapolated preglacial temperature of 11°C is consistent with a relative temperature difference of 26°C between preglacial and glacial temperatures reconstructed from clumped oxygen isotopes (authors' Ref. 25), and consistent with a preglacial temperature of 9°C from ikaite (authors' Ref. 26, 27). The reviewers feel that this part of the discussion does not adequately consider the broader range of environmental interpretations possible when the 95% confidence limits of the current study (yielding a range of sea water temperatures of -22°C to -8°C) are taken into consideration. Several questions arise:

- What is the basis for extrapolating a reconstructed preglacial temperature of 11°C ?
- Mackey et al. (authors' Ref. 25) concluded that "[m]ean glacial autochthonous dolomitic $\Delta 47$ temperatures are $26 \pm 10^{\circ}\text{C}$ (95% CL) cooler than preglacial strata" and that "the preserved temperature and $\delta 18\text{O}$ differences between low-latitude preglacial Tonian and synglacial Cryogenian dolomites are an imperfect reflection of primary temperature change and ice sheet expansion" owing to the influence of early diagenesis on preserved $\delta 18\text{O}$ trends. Field et al (authors' Ref. 27) notes that the 9°C temperature associated with ikaite deposition is a maximum value, and has been more typically measured at $<6^{\circ}\text{C}$. What are the ramifications for the current study's conclusions based on a reconstructed sea water temperature range of $-15^{\circ}\text{C} \pm 7^{\circ}\text{C}$ (the full range is important!) if the broader range of temperature constraints mentioned in Refs. 25-27 are taken into consideration?
- The preglacial units cited in authors' Refs. 25 and 26 are roughly contemporaneous in age and are both interpreted as low-latitude deposits, but their respective sea water temperature interpretations differ: Mackey et al. (authors' Ref. 25) infer a warm preglacial period (approx. 30°C warmer than "the glacial average," which is not given a specific temperature range owing to inferred diagenetic effects), whereas Trower et al (authors' Ref. 26) conclude that the preglacial interval was already cool leading into the Cryogenian ($\sim 9^{\circ}\text{C}$). In order to best utilize these studies as supporting evidence for the current work, how would the authors reconcile these differences in interpretation?

High salinity ramifications. The authors argue (lines 209-221) that a salinity of ~ 150 psu corresponding to sea water temperature of at least -11°C (lines 143-147) and thicker sea ice (lines 193-194) implies a larger change in relative sea level than previously thought. If the global ocean had a salinity of ~ 150 psu, a simple back-of-the-envelope calculation suggests that would require a $>75\%$ reduction in volume compared to the modern ocean. Such a state should yield testable hypotheses regarding the large-scale erosion of continental margins and severe environmental impacts on life in the oceans at that time, and arguably, such impacts are at least as significant as an analytical finding of a sub-zero global ocean. However, the authors appear to reject significant glacial erosion for other reasons (lines 76-84), and do not address the significant challenges that high salinity would pose to life forms already impacted by extreme cold and (for photosynthetic organisms) limited light availability. The extent to which the current study's extraordinary conclusions poses challenges to other lines of environmental interpretation should be presented in some greater detail, even if resolving any differences in interpretation are beyond the scope of the current manuscript.

Additional comments:

Lines 57-58: What are the bounds to a "wide range of variability" in $\delta 56\text{Fe}$ values as shown in Fig 1? While the range of values circa 2400 Ma is indeed broad (approx. 4.5 per mil), the time slice at ~ 700 Ma has roughly the same range (approx. 4.5 per mil), and the time slices at ~ 3000 Ma (range approx. 3.25 per mil) and ~ 1700 Ma (range approx. 2.9 per mil) are notably broader than the remaining ones shown.

Fig. 2: We are not convinced that the interpretation of this plot as indicating a lack of secular change in Fe sources to the oceans over 4000 Ma of Earth history is correct. The green line suggests a baseline $\delta 56\text{Fe}$ value of ~ 0.1 per mil, which appears a bit low given that the data points for many of the time slices skew more positively than this baseline value. It would be helpful to see the slope equation of the green line as well as a statistical analysis comparing the slope of the line to a line with slope = 0.

Lines 80-81: The statement, "However, firstly, the Fe isotopes of dissolved iron in modern East Antarctic glacial meltwater is strongly negative[16]," does not track as a rebuttal to the hypothesis that the unusually isotopically heavy Fe in CIF could have been the result glacial erosion. East Antarctica has a much different geologic history compared to the Cryogenian landscape, and therefore the negative Fe isotopes of Taylor Glacier's subglacial waters, as described in authors' Ref. 16, are likely not an appropriate contrast to the Cryogenian results.

Fig. 3a: The scale of the y axis of this figure makes interpreting the differences between the preGOE and Cryogenian distributions difficult. It would be helpful to include a simple statistical statement in the caption to assure readers that the difference is significant.

Lines 136-137: The authors argue that "strong support for a salinity-controlled freezing point comes from extremely cold temperatures of -13°C recorded in analogous Antarctic brines (Ref. 31)." The geographic context of this -13°C temperature

(authors' Ref. 31) is a buried Antarctic lake -- Lake Vida -- with salinity of 200 psu. The reviewers would argue that, from the standpoint of salinity and sea water temperature, citing Lake Vida as an analogous environment is supporting evidence for the CIF depositional settings as restricted basins, with no relation to global ocean conditions.

Figure 4: The presentation of salinity in the plot would benefit from a second y axis indicating salinity as a psu value, for ease of comparison to the text discussion of salinity in terms of psu.

Lines 194-195: The manuscript cites Tziperman et al [Ref. 40] as an example of a global circulation model that predicted the Snowball global ocean was covered with ~1-km-thick sea ice, and the authors suggest this may represent an underestimate given the reconstructed sea water temperatures of this study. However, Tziperman et al. utilized a two-dimensional global ice flow model that assumes at the outset that sea ice must be very thick, and Tziperman et al. concede that their model "ignores many known important factors such as thermodynamic, optical effects, dust and dust transport, and is therefore meant as a process study focusing on one specific effect, rather than a realistic simulation of Neoproterozoic ice thickness." To date, these reviewers are not aware of any three-dimensional global climate model -- which incorporates all aspects of Earth's climate system (not just sea ice) -- that has developed ~1 km-thick sea ice as an emergent response to snowball-type climate forcings. Note e.g. that Liu et al (authors' Ref. 52) do not indicate more than ~20-26 m total sea ice thickness for Snowball Earth climate simulations using the CCSM3 global climate model (their Figs. 6, 13a and A2).

Supplementary Fig. 3: It seems as if the signs of the x axis are reversed. In figure 1 of the paper, the Cryogenian samples have a positive skew, but in this figure they have a negative skew.

Reviewer #4

(Remarks to the Author)

Version 2:

Reviewer comments:

Reviewer #3

(Remarks to the Author)

Reviewers' comments on second revision of manuscript now entitled: "Extremely cold ocean temperatures in iron formation brine pools of snowball Earth" by Lu et al. The authors have made extensive revisions in response to previous comments, and the latest revision of this manuscript presents a more defensible interpretation of their temperature-independent iron isotope results. However, we do have some additional comments.

Line numbers refer to the second revised manuscript with integrated edits.

Sea water temperatures. The authors have added (lines 159-166) some additional text and a new reference to expand upon the point previously made, about the consistency between Mackey et al.'s (authors' ref. 26) preglacial and glacial temperature difference in relative terms, and a preglacial and glacial temperature difference based on the author's reconstruction and a formation temperature of 9°C of ikaite. While this clarifying language and the new reference are welcome, the authors did not quite address our previous comment suggesting that the full range of possible interpretations need to be addressed. To be more explicit:

Line 161 mentions that the relative temperature difference identified by Mackey et al. is 26°C (no ± error stated). However, Mackey et al. described this difference as 26°C ± 10°C. Given that the typical formation temperature of ikaite in nature is <6°C (authors' ref. 28), the authors' analogy holds best as long as Mackey et al.'s relative temperature difference falls somewhere between 16 and 21°C (to accommodate a ≤6°C formation temperature of ikaite) and the authors' reconstructed temperature lies between -22°C and -10°C.

We suggest, in the interest of full transparency, that the Mackey et al. error bounds are made explicit in the discussion. This would not detract from the authors' argument, and could in fact be of interest as an additional constraint to Mackey et al.

Sea ice thickness. The authors state that increasing δ⁵⁶Fe isotopes upsection provides "strong support for a 'hard' snowball Earth" (line 322), and that the general occurrence of CIF at low latitudes is consistent with "their isotopic signature of thick sea ice" (line 324) that is "incompatible with 'slushball' or 'waterbelt' solutions involving open ocean in equatorial regions with high sublimation" (lines 325-326). Given that "CIF are generally thought to have been deposited in semi-restricted basins" (lines 225-226), and the δ⁵⁶Fe isotopes are a proposed proxy for sea water temperature and not sea ice thickness, we find that these closing lines make broad statements about the temperature and sea ice cover of the tropical ocean away from continental margins that are not supported by the information provided in the current study. We contend that the use of "strong" and "incompatible" are not appropriate terms considering the limited scope of the evidence provided. We suggest the use of less definitive language. In our opinion the evidence i) "provides support for a 'hard' snowball Earth" and ii) is unable to address the presence (or absence) of a 'slushball' or 'waterbelt'.

Additional comments:

Fig. 5: Typo in the figure labeling, should read “-9°C, McMurdo brines”

Fig 5 caption: The authors refer to a contrast between the reconstructed temperatures of the current study and an ambient snowball ocean temperature of -5.5°C, authors' ref. 51 (lines 264-265). Since ref. 51 is not yet publicly available, it is unclear what the source of the -5.5°C value is, and whether that refers to an average temperature of the entire global ocean or just some portion of it (e.g., upper ocean). Further information is necessary to evaluate this statement.

Reviewer #4

(Remarks to the Author)

Responses in blue

Revisions in red

Reviewers' comments:

Reviewer #1 (Remarks to the Author):

I feel that the authors have done a good job at addressing my concerns. Although I would have liked to see bit more details about how the surges of oxidized glacial meltwater worked, it is a fair point that this topic has already been covered by the co-authors elsewhere. Thus, I am happy to accept the manuscript.

Response: We greatly appreciate the reviewer's repeated consideration of our manuscript and their positive assessment of it this last round.

Revision: To address the reviewer's additional request, and in light of comments by new Reviewers 3/4, we added a new final figure that should also help elucidate the meltwater discharge model that Reviewer #1 also hoped to see more detail on. Details provided later below.

Reviewer #2 (Remarks to the Author):

This ms presents a very interesting observation that Fe isotope compositions of the Cryogenian BIF are among the heaviest in Earth's history. As isotope fractionation is negatively correlated with temperature in general, this observation would imply that the temperature for Cryogenic BIF is extremely low, and this is what are the authors meant. However, BIF precipitates from water, and the freezing point of water is almost constant. Therefore, the fundamental reasoning of the ms is wrong, thus rejection is strongly suggested.

Response: The statement that "the freezing pint of water is almost constant" is incorrect with regards to basic thermodynamics. Salinity can significantly reduce freezing point. As we have not only demonstrated temperature-dependent fractionation suggesting very negative ocean temperatures, but also demonstrated strongly elevated salinity that such a cold freezing temperature would require, our two findings are internally consistent, not to mention consistent with one might expect in a Cryogenian iron formation brine pool (see revisions based on Reviewers 3/4 below).

Reviewer #3 (Remarks to the Author):

Reviewers' comments for revised manuscript entitled: "Extremely cold ocean temperatures during snowball Earth" by Lu et al. This study describes the use of temperature-dependent iron isotope fractionation of Cryogenian iron formations (CIF) to propose that the temperature of the Snowball oceans was $-15^{\circ}\text{C} \pm 7^{\circ}\text{C}$, and that marine ice was correspondingly very thick. The

application of temperature-dependent Fe fractionation to ocean temperature reconstructions is of interest, especially for time periods when many of the proxies available for more recent Earth history cannot be applied. At the request of the Editor, we focused on the uncertainty and plausibility of the revised manuscript's temperature reconstruction, and the authors' responses to previous reviewers' remarks.

We disagree with the manuscript's overall conclusion that the reconstructed ocean temperatures apply to the global Cryogenian ocean. We do not dispute the temperature reconstructions; rather, we find that the framing of the environmental context of these results, and the subsequent ramifications for global snowball conditions, are not adequately supported by the arguments given. The claims made in this manuscript would benefit from greater exposition, perhaps in another journal conducive to longer manuscripts as suggested by previous Reviewer #2.

Line numbers refer to the revised manuscript with integrated edits.

CIF depositional settings. The authors note briefly (in lines 202-204) that CIF are associated with semi-restricted basins that may not have had a substantial connection to the open snowball ocean. The reviewers feel that it is more likely that the reconstructed temperatures are in fact capturing conditions in restricted basins. Note that the Holowilena Formation is found within Sturtian-age rift basins of the Adelaide Superbasin (see e.g. authors' Ref. 17; Lloyd et al., Geological Magazine (2023)), and Neoproterozoic IF in general are typically associated with rift settings (authors' Ref. 13). The authors dismiss a scenario in which the waters of these basins may have become isolated from the global ocean by the overflow of land-based glaciers, leading to the significantly higher salinity suggested by the Sr/Ba data (authors' Ref. 5). The authors deem the isolation of Holowilena unlikely owing to the presence of submarine ridge volcanism (authors' Ref. 46). However, none of the large scale volcanic events examined in authors' Ref. 46 are in the geographic vicinity of Holowilena deposition. Furthermore, the authors have not addressed the possibility that smaller-scale hydrothermal vents supplying Fe could have been active within these rift basins, as they are e.g. in the modern Red Sea (van der Zwan et al., Communications Earth & Environment (2023)). Considering this omission of a more in-depth discussion, we argue that the authors do not sufficiently support their assertion that their findings are representative of the global ocean.

Revision: We thank Reviewers 3/4 for helping us realize—actually in line with much of the past work of some of our co-authors—that the CIF $\delta^{56}\text{Fe}$ temperature and salinity estimates are *not* representative of the global ocean, but rather a globally distributed CIF depositional setting, including the following criteria:

- (i) Near ice grounding line
- (ii) Restriction
- (iii) Brine pools

This critical revision greatly increases the specificity and certainty of our result, and provides an internally consistent result for both the extreme cold and extreme salinity of the CIF during snowball Earth. Thus, our method+result remained unchanged, but the context has modified the

implications, which still offer critical insights into glaciomarine settings distributed globally during snowball Earth and the unique reprisal of iron formation. Here are some of the revisions that have been made to address this major revision:

- (i) The title has been modified (**bold**) to “Extremely cold ocean temperatures **in iron formation brine pools** of snowball Earth”
- (ii) Both the Abstract *setup* and *finish* have been modified (**bold**):
Setup: “However, **cold, salty**, ice-covered oceans—a salient prediction of snowball Earth—is difficult to establish geologically.”
Finish: “...we calculate that the temperature **of the iron formation brine pools** was $-15 \pm 7^\circ\text{C}$. Such cold **snowball brine pools, colder than those in Antarctic margins today, represent** Earth’s coldest recorded ocean temperatures.”
- (iii) A new Figure 1 now shows an updated paleogeographic reconstruction showing the globally distributed occurrences of CIF, underlying that while our result is not emblematic of the global ocean, it represents a globally significant depositional setting during the snowball Earth, with implications for potential refugia for the survival of life during extreme glaciation.
- (iv) The Discussion has essentially been nearly entirely rewritten to accommodate this slight, but not subtle, shift in the scope of our interpretation of the $\delta^{56}\text{Fe}$ -based T estimate.
- (v) Previous Figure 5—comparison with oxygen-isotope-based surface T estimates—which was only referred to in one sentence in the text, and not commented on by Reviewers 3/4, was removed. It did not contribute too much additional support and, since adding a new final figure on brines and a CIF brine production model for snowball, we thought it would be better not to distract from the main message.
- (vi) Instead, a new final figure (Fig. 6) has been added that drives home the new take-home message that the $\delta^{56}\text{Fe}$ -based T estimate of seawater relates to iron formation brine pools. Figure 6a compares the snowball T estimates and modern ones from Antarctic brines. Figure 6b provides a geologic model for how such cold, salty snowball brines might have been developed in Cryogenian iron formation depositional settings.

We cannot thank the reviewers enough for being firm on what our result does and doesn’t mean. We hope you agree that the substantially revised, and much more focused manuscript still provides an innovative geochemical method and a geologically important result for better understanding the severe conditions of Cryogenian climate as viewed through its unique iron formation.

Sea water temperatures. The authors argue (lines 116-124) that the difference between their calculated glacial sea water temperature of -15°C and an extrapolated preglacial temperature of 11°C is consistent with a relative temperature difference of 26°C between preglacial and glacial temperatures reconstructed from clumped oxygen isotopes (authors’ Ref. 25), and consistent with a preglacial temperature of 9°C from ikaite (authors’ Ref. 26, 27). The reviewers feel that this part of the discussion does not adequately consider the broader range of environmental interpretations possible when the 95% confidence limits of the current study (yielding a range of sea water temperatures of -22°C to -8°C) are taken into consideration. Several questions arise:

- What is the basis for extrapolating a reconstructed preglacial temperature of 11°C ?

Response: We believe this is already explained quite clearly in the main text:

“Although the *absolute* temperatures are uncertain in this case, the *relative* temperature difference between glacial and preglacial temperatures is determinable and is 26 °C (Mackey+20_AGUAdv¹). Thus, if the temperature of the seawater during the Sturtian snowball was exactly –15 °C as we have calculated, then the extrapolated pre-glacial temperature would have been 11 °C...”

$$-15^{\circ}\text{C} + 26^{\circ}\text{C} = 11^{\circ}\text{C}$$

Revision: The text has nonetheless been slightly modified to make this even more clear. A since-published study corroborating, independently of ikaite, a cold climate before snowball (Trower+25_GEO²) has also been added.

- Mackey et al. (authors' Ref. 25) concluded that “[m]ean glacial autochthonous dolomicrite $\Delta 47$ temperatures are $26 \pm 10^{\circ}\text{C}$ (95% CL) cooler than preglacial strata” and that “the preserved temperature and $\delta 18\text{O}$ differences between low-latitude preglacial Tonian and synglacial Cryogenian dolomites are an imperfect reflection of primary temperature change and ice sheet expansion” owing to the influence of early diagenesis on preserved $\delta 18\text{O}$ trends. Field et al (authors' Ref. 27) notes that the 9°C temperature associated with ikaite deposition is a maximum value, and has been more typically measured at $<6^{\circ}\text{C}$. What are the ramifications for the current study's conclusions based on a reconstructed sea water temperature range of $-15^{\circ}\text{C} \pm 7^{\circ}\text{C}$ (the full range is important!) if the broader range of temperature constraints mentioned in Refs. 25-27 are taken into consideration?

- The preglacial units cited in authors' Refs. 25 and 26 are roughly contemporaneous in age and are both interpreted as low-latitude deposits, but their respective sea water temperature interpretations differ: Mackey et al. (authors' Ref. 25) infer a warm preglacial period (approx. 30°C warmer than “the glacial average,” which is not given a specific temperature range owing to inferred diagenetic effects), whereas Trower et al (authors' Ref. 26) conclude that the preglacial interval was already cool leading into the Cryogenian ($\sim 9^{\circ}\text{C}$). In order to best utilize these studies as supporting evidence for the current work, how would the authors reconcile these differences in interpretation?

Response: Clumped isotope thermometry faces significant challenges in reconstructing original water temperatures during carbonate deposition due to pervasive diagenetic alteration. However, the paleogeographic context provides a critical constraint: both Svalbard and Death Valley occupied proximal low-latitude positions before, during, and after the Sturtian glaciation. Leveraging this spatial and temporal coherence, we utilize two key parameters:

- (1) A documented 26°C temperature difference between these coeval sites under a shared thermal history (McKay+23_AGUAdv¹), and
- (2) The presence of pre-glacial ikaite. This mineral's stability constrains maximum water temperatures in these low-latitude regions to $\leq 9^{\circ}\text{C}$, with typical conditions below 6°C (Field+17_MinMag³).

Integrating these constraints permits estimation of glacial carbonate deposition temperatures. Calculations suggest values as low as -17°C , potentially reaching -20°C . Notably, this estimated range (-17°C to -20°C) falls within the independently derived low-latitude aquatic temperature envelope of -8°C to -22°C established through our iron isotope proxy.

High salinity ramifications. The authors argue (lines 209-221) that a salinity of ~ 150 psu corresponding to sea water temperature of at least -11°C (lines 143-147) and thicker sea ice (lines 193-194) implies a larger change in relative sea level than previously thought. If the global ocean had a salinity of ~ 150 psu, a simple back-of-the-envelope calculation suggests that would require a $>75\%$ reduction in volume compared to the modern ocean. Such a state should yield testable hypotheses regarding the large-scale erosion of continental margins and severe environmental impacts on life in the oceans at that time, and arguably, such impacts are at least as significant as an analytical finding of a sub-zero global ocean. However, the authors appear to reject significant glacial erosion for other reasons (lines 76-84), and do not address the significant challenges that high salinity would pose to life forms already impacted by extreme cold and (for photosynthetic organisms) limited light availability. The extent to which the current study's extraordinary conclusions poses challenges to other lines of environmental interpretation should be presented in some greater detail, even if resolving any differences in interpretation are beyond the scope of the current manuscript.

Revision: We thank Reviewers 3/4 again for pointing out that the depositional context of CIF is not only important for interpreting their extremely cold temperatures, but also their extremely high salinity. That is, modifying the interpretation from the global ocean to the globally distributed CIF ice-grounding line brine pools resolves the certainty and nuance of our result. Please note that the Discussion has essentially been nearly entirely rewritten to accommodate this slight, but not subtle, shift in the scope of our interpretation of the $\delta^{56}\text{Fe}$ -based T estimate. Specifically here, too, we appreciated the point about the implied reduction in ocean volume that provides an important prediction to consider—which we have incorporated essentially verbatim, thanks. We also have addressed how modern analogs for CIF of sub-ice shelf Antarctic brines help understand the severe cold and salinity, as well as not precluding such a setting as a refugium for at least microbial life during severe snowball conditions. Thanks for *all* of the suggestions, most of which have been incorporated in the revision.

Additional comments:

Lines 57-58: What are the bounds to a “wide range of variability” in $\delta^{56}\text{Fe}$ values as shown in Fig 1? While the range of values circa 2400 Ma is indeed broad (approx. 4.5 per mil), the time slice at ~ 700 Ma has roughly the same range (approx. 4.5 per mil), and the time slices at ~ 3000 Ma (range approx. 3.25 per mil) and ~ 1700 Ma (range approx. 2.9 per mil) are notably broader than the remaining ones shown.

Response: With all due respect to the reviewers, we stand by this sentence as written. Perhaps the reviewers misread it, as the CIF interval *is* to ~ 700 Ma timescale; and this sentence is pointing out

exactly what the reviewers noted: the comparable large variability of the ca. 2400 and 700 Ma time slices. (Meanwhile, the variability at 1700 Ma is not nearly as large and relies on only 3 [arguably outlier] data, unlike the aforementioned time slices.)

Fig. 2: We are not convinced that the interpretation of this plot as indicating a lack of secular change in Fe sources to the oceans over 4000 Ma of Earth history is correct. The green line suggests a baseline $\delta^{56}\text{Fe}$ value of ~ 0.1 per mil, which appears a bit low given that the data points for many of the time slices skew more positively than this baseline value. It would be helpful to see the slope equation of the green line as well as a statistical analysis comparing the slope of the line to a line with slope = 0.

Response: While we appreciate the suggestion to compare this line with free slope to a line with slope = 0, we have already done so, and double-checked again, and the two lines are visually indistinguishable from each other.

Response: Furthermore, to make this point more quantitatively clear, we have added the miniscule slope of the free-slope line to the figure caption, which is $m = 5 \times 10^{-14}$, or effectively zero.

Nonetheless, in light of the reviewers saying not finding this figure terribly convincing for the point it is used to make, we have opted to move it back into the supplementary information (from whence it came)—also, since we have added two new figures to address other more salient points.

Lines 80-81: The statement, “However, firstly, the Fe isotopes of dissolved iron in modern East Antarctic glacial meltwater is strongly negative[16],” does not track as a rebuttal to the hypothesis that the unusually isotopically heavy Fe in CIF could have been the result glacial erosion. East Antarctica has a much different geologic history compared to the Cryogenian landscape, and therefore the negative Fe isotopes of Taylor Glacier’s subglacial waters, as described in authors’ Ref. 16, are likely not an appropriate contrast to the Cryogenian results.

Revision: Although we may disagree that Taylor glacier is not a bad analog for Cryogenian, since we already have another line of argument here, we have just removed this disputed one.

Fig. 3a: The scale of the y axis of this figure makes interpreting the differences between the preGOE and Cryogenian distributions difficult. It would be helpful to include a simple statistical statement in the caption to assure readers that the difference is significant.

Response: Excellent suggestion. The following has been added to the figure caption:

“A Mood test for equal medians yields a p value of 8.5×10^{-9} , which is $\lll 0.05$, indicating that the null hypothesis can be rejected and that the two medians are strongly statistically distinct.”

Lines 136-137: The authors argue that “strong support for a salinity-controlled freezing point comes from extremely cold temperatures of -13°C recorded in analogous Antarctic brines (Ref. 31).” The geographic context of this -13°C temperature (authors’ Ref. 31) is a buried Antarctic

lake -- Lake Vida -- with salinity of 200 psu. The reviewers would argue that, from the standpoint of salinity and sea water temperature, citing Lake Vida as an analogous environment is supporting evidence for the CIF depositional settings as restricted basins, with no relation to global ocean conditions.

Response: Many thanks, again, for making this point because it has pushed us to critically revise the context and main implication of our method+result, shifting from our previous argument of the “global ocean” to now the “globally distributed CIF snowball brine pools.” As detailed elsewhere, this revision is expressed throughout the revised manuscript in a multitude of edits that improve its accuracy and therefore also its certainty. Please note that the Discussion has essentially been nearly entirely rewritten to accommodate this slight, but not subtle, shift in the scope of our interpretation of the $d^{56}\text{Fe}$ -based T estimate.

Figure 4: The presentation of salinity in the plot would benefit from a second y axis indicating salinity as a psu value, for ease of comparison to the text discussion of salinity in terms of psu.

Response: While we appreciate the suggestion, and would enact it if we could, there is no direct conversion available for converting Sr/Ba ratio proxy to salinity (psu units). This nonetheless does not detract from our ability to compare the ~4 times difference in Sr/Ba CIF ratios with those in the modern ocean to yield at least a broad estimate of CIF brine pool salinity.

Revision: But, yes, this section on salinity has been *clearly* revised in order to focus on CIF brine pools during snowball and *not* the global snowball ocean.

Lines 194-195: The manuscript cites Tziperman et al [Ref. 40] as an example of a global circulation model that predicted the Snowball global ocean was covered with ~1-km-thick sea ice, and the authors suggest this may represent an underestimate given the reconstructed sea water temperatures of this study. However, Tziperman et al. utilized a two-dimensional global ice flow model that assumes at the outset that sea ice must be very thick, and Tziperman et al. concede that their model “ignores many known important factors such as thermodynamic, optical effects, dust and dust transport, and is therefore meant as a process study focusing on one specific effect, rather than a realistic simulation of Neoproterozoic ice thickness.” To date, these reviewers are not aware of any three-dimensional global climate model -- which incorporates all aspects of Earth’s climate system (not just sea ice) -- that has developed ~1 km-thick sea ice as an emergent response to snowball-type climate forcings. Note e.g. that Liu et al (authors’ Ref. 52) do not indicate more than ~20-26 m total sea ice thickness for Snowball Earth climate simulations using the CCSM3 global climate model (their Figs. 6, 13a and A2).

Revision: We appreciate the reviewers’ knowledge of snowball sea-ice thickness models. As the focus of this paper—thanks to your insightful feedback—has been redirected away from “global ocean” implications, such text has been altogether removed or revised with these comments in mind. Please note that the Discussion has essentially been nearly entirely rewritten to accommodate this slight, but not subtle, shift in the scope of our interpretation of the $d^{56}\text{Fe}$ -based T estimate.

Supplementary Fig. 3: It seems as if the signs of the x axis are reversed. In figure 1 of the paper, the Cryogenian samples have a positive skew, but in this figure they have a negative skew.

Revision: Hawkeye. Thanks for the attention to detail. Indeed, the signs of the x-axis were reversed and have been corrected.

Reviewer #4 (Remarks to the Author):

Response: We commend both *Nature Communications* for such a wise program and you (and your mentor) for encouraging the art of (helpful!) critical peer review early on.

References

- 1 Mackey, T. J., Jost, A. B., Creveling, J. R. & Bergmann, K. D. A Decrease to Low Carbonate Clumped Isotope Temperatures in Cryogenian Strata. *AGU Advances* **1**, e2019AV000159 (2020). <https://doi.org/https://doi.org/10.1029/2019AV000159>
- 2 Trower, E. J., Ingalls, M., Gutoski, J. R. & Wala, V. T. New constraints on phosphate concentration and temperature in shallow late Tonian seawater. *Geology* (2025).
- 3 Field, L. P. *et al.* Unusual morphologies and the occurrence of pseudomorphs after ikaite (CaCO₃·6H₂O) in fast growing, hyperalkaline speleothems. *Mineralogical Magazine* **81**, 565-589 (2017). <https://doi.org/10.1180/minmag.2016.080.111>

Revisions are in red.

Reviewer #3 (Remarks to the Author)

Reviewers' comments on second revision of manuscript now entitled: "Extremely cold ocean temperatures in iron formation brine pools of snowball Earth" by Lu et al. The authors have made extensive revisions in response to previous comments, and the latest revision of this manuscript presents a more defensible interpretation of their temperature-independent iron isotope results. However, we do have some additional comments.

Line numbers refer to the second revised manuscript with integrated edits.

Sea water temperatures. The authors have added (lines 159-166) some additional text and a new reference to expand upon the point previously made, about the consistency between Mackey et al.'s (authors' ref. 26) preglacial and glacial temperature difference in relative terms, and a preglacial and glacial temperature difference based on the author's reconstruction and a formation temperature of 9 °C of ikaite. While this clarifying language and the new reference are welcome, the authors did not quite address our previous comment suggesting that the full range of possible interpretations need to be addressed. To be more explicit:

Line 161 mentions that the relative temperature difference identified by Mackey et al. is 26 °C (no \pm error stated). However, Mackey et al. described this difference as 26 °C \pm 10 °C. Given that the typical formation temperature of ikaite in nature is <6 °C (authors' ref. 28), the authors' analogy holds best as long as Mackey et al.'s relative temperature difference falls somewhere between 16 and 21 °C (to accommodate a ≤ 6 °C formation temperature of ikaite) and the authors' reconstructed temperature lies between -22 °C and -10 °C.

We suggest, in the interest of full transparency, that the Mackey et al. error bounds are made explicit in the discussion. This would not detract from the authors' argument, and could in fact be of interest as an additional constraint to Mackey et al.

Revision: Excellent suggestion. We will incorporate the error ($\pm 10^\circ\text{C}$) you mentioned. Furthermore, if we take this error into account, along with the formation temperature of ikaite being $\leq 6^\circ\text{C}$, our reconstructed glacial seawater temperature would still range from -10°C to -30°C . This is consistent with our estimate of glacial seawater temperature derived from iron isotopes, which falls between -8°C and -22°C . This qualification has been added to the revised text.

Sea ice thickness. The authors state that increasing $\delta^{56}\text{Fe}$ isotopes upsection provides "strong support for a 'hard' snowball Earth" (line 322), and that the general occurrence of CIF at low latitudes is consistent with "their isotopic signature of thick sea ice" (line 324) that is "incompatible with 'slushball' or 'waterbelt'

solutions involving open ocean in equatorial regions with high sublimation” (lines 325-326). Given that “CIF are generally thought to have been deposited in semi-restricted basins” (lines 225-226), and the $\delta^{56}\text{Fe}$ isotopes are a proposed proxy for sea water temperature and not sea ice thickness, we find that these closing lines make broad statements about the temperature and sea ice cover of the tropical ocean away from continental margins that are not supported by the information provided in the current study. We contend that the use of “strong” and “incompatible” are not appropriate terms considering the limited scope of the evidence provided. We suggest the use of less definitive language. In our opinion the evidence i) “provides support for a ‘hard’ snowball Earth” and ii) is unable to address the presence (or absence) of a ‘slushball’ or ‘waterbelt’ .

Revision: Fair point. Point taken, and excellent suggestion for revision. The final two lines of the manuscript have been revised accordingly:

“Increasing magnetic susceptibility associated with an increasing abundance of hematite that corresponds closely with increasing $\delta^{56}\text{Fe}$ values upsection¹ can therefore be attributed with colder temperatures of glacial advance. This collective observation provides support for a “hard” snowball Earth, where oxidation for CIF deposition only occurred from surges of oxidized continental-derived meltwater², but is unable to address the presence (or absence) of a “soft” snowball or ‘waterbelt’^{3,4}.”

Additional comments:

Fig. 5: Typo in the figure labeling, should read “-9 C, McMurdo brines”

Revision: Good eye. Fixed.

Fig 5 caption: The authors refer to a contrast between the reconstructed temperatures of the current study and an ambient snowball ocean temperature of -5.5 C, authors’ ref. 51 (lines 264-265). Since ref. 51 is not yet publicly available, it is unclear what the source of the -5.5 C value is, and whether that refers to an average temperature of the entire global ocean or just some portion of it (e.g., upper ocean). Further information is necessary to evaluate this statement.

Revision: Good point and smart request. Both points of information are now provided in this addition to the caption, as well as an additional reference that is key to the Hoffman textbook calculation:

“Estimate for global snowball ocean is based on slight pressure-based freezing-point depression from the overlying sea glacier, under a range of salinities and water depths (or equivalent ice thicknesses)^{5,6}.”

Reviewer #4 (Remarks to the Author)

- 1 Mitchell, R. N. *et al.* Orbital forcing of ice sheets during snowball Earth. *Nature Communications* **12**, 4187 (2021). <https://doi.org/10.1038/s41467-021-24439-4>
- 2 Lechte, M. A. *et al.* Subglacial meltwater supported aerobic marine habitats during Snowball Earth. *Proceedings of the National Academy of Sciences* **116**, 25478-25483 (2019).
- 3 Hyde, W. T., Crowley, T. J., Baum, S. K. & Peltier, W. R. Neoproterozoic 'snowball Earth' simulations with a coupled climate/ice-sheet model. *Nature* **405**, 425-429 (2000).
- 4 Abbot, D. S., Voigt, A. & Koll, D. The Jormungand global climate state and implications for Neoproterozoic glaciations. *Journal of Geophysical Research* **116**, D18103 (2011).
- 5 Hoffman, P. F. *Snowball Earth in 4-G: Geology, Geophysics, Geochemistry, Geobiology*. (Phoenix Science Press, in press).
- 6 Feistel, R. & Marion, G. M. A Gibbs–Pitzer function for high-salinity seawater thermodynamics. *Progress in Oceanography* **74**, 515-539 (2007). <https://doi.org/https://doi.org/10.1016/j.pocean.2007.04.020>